# Antiangiogenic Properties of Axitinib versus Sorafenib Following Sunitinib Resistance in Human Endothelial Cells—A View towards Second Line Renal Cell Carcinoma Treatment

**DOI:** 10.3390/biomedicines9111630

**Published:** 2021-11-06

**Authors:** Eva Juengel, Pascal Schnalke, Jochen Rutz, Sebastian Maxeiner, Felix K.-H. Chun, Roman A. Blaheta

**Affiliations:** 1Department of Urology, Goethe-University, 60590 Frankfurt am Main, Germany; Eva.Juengel@unimedizin-mainz.de (E.J.); pascal.schnalke@gmail.com (P.S.); Jochen.Rutz@kgu.de (J.R.); Sebastian.Maxeiner@kgu.de (S.M.); Felix.Chun@kgu.de (F.K.-H.C.); 2Department of Urology and Pediatric Urology, University Medical Center Mainz, 55131 Mainz, Germany

**Keywords:** renal cell carcinoma, tyrosine kinase inhibitors, endothelial cells, resistance, second-line

## Abstract

Tyrosine kinase inhibitors (TKIs) and immune checkpoint inhibitors predominate as first-line therapy options for renal cell carcinoma. When first-line TKI therapy fails due to resistance development, an optimal second-line therapy has not yet been established. The present investigation is directed towards comparing the anti-angiogenic properties of the TKIs, sorafenib and axitinib on human endothelial cells (HUVECs) with acquired resistance towards the TKI sunitinib. HUVECs were driven to resistance by continuously exposing them to sunitinib for six weeks. They were then switched to a 24 h or further six weeks treatment with sorafenib or axitinib. HUVEC growth, as well as angiogenesis (tube formation and scratch wound assay), were evaluated. Cell cycle proteins of the CDK-cyclin axis (CDK1 and 2, total and phosphorylated, cyclin A and B) and the mTOR pathway (AKT, total and phosphorylated) were also assessed. Axitinib (but not sorafenib) significantly suppressed growth of sunitinib-resistant HUVECs when they were exposed for six weeks. This axinitib-associated growth reduction was accompanied by a cell cycle block at the G0/G1-phase. Both axitinib and sorafenib reduced HUVEC tube length and prevented wound closure (sorafenib > axitinib) when applied to sunitinib-resistant HUVECs for six weeks. Protein analysis revealed diminished phosphorylation of CDK1, CDK2 and pAKT, accompanied by a suppression of cyclin A and B. Both drugs modulated CDK-cyclin and AKT-dependent signaling, associated either with both HUVEC growth and angiogenesis (axitinib) or angiogenesis alone (sorafenib). Axitinib and sorafenib may be equally applicable as second line treatment options, following sunitinib resistance.

## 1. Introduction

About 3% of all malignancies worldwide are related to renal carcinomas, with renal cell carcinoma (RCC) comprising about 90–95% of all renal cancer [1]. One third of RCC patients already have metastases at diagnosis, and a further 30–70% of patients with localized disease relapse following initial surgery [2]. During the last years, novel drugs and drug combinations have been approved, considerably widening therapeutic options.

Multiple tyrosine kinase inhibitors (TKIs), acting on specific growth factor receptors or drugs interfering with the mechanistic target of rapamycin (mTOR) pathway have shown promise in terms of progression free and overall survival [3]. Sorafenib was the first multi-targeted TKI approved for first-line treatment of metastatic RCC, which was then replaced by sunitinib and pazopanib as the standard of care. Recently, the NCCN Guidelines has recommended axitinib as the first-line option for clear cell RCC with favorable and poor/intermediate risk and cabozantinib for patients with poor/intermediate risk. The combination of lenvatinib and the mTOR-inhibitor, everolimus, is recommended for the subsequent therapy of clear cell RCC and systemic therapy for non-clear cell RCC [4].

However, none of these regimens are curative, with resistance inevitably developing during therapy. Novel concepts concentrating on immune checkpoint inhibitors (ICI), targeting programmed death 1 (PD-1), the PD-Ligand 1 (PD-L1) and cytotoxic T-lymphocyte–associated antigen 4 (CTLA-4), are meanwhile under investigation. ICIs include nivolumab (anti–PD-1), pembrolizumab (anti–PD-1), avelumab (anti–PD-L1), atezolizumab (anti–PD-L1) and ipilimumab (anti–CTLA-4) [5]. Currently, the recommended first-line treatment for metastatic RCC patients with intermediate and poor-risk features consists of a combination of an ICI plus a TKI [6]. Consensus as to second-line treatment has not been reached and therapy remains more empirical than evidence based [7]. Based on the European Association of Urology (EAU), the European Society for Medical Oncology (ESMO) and the National Comprehensive Cancer Network (NCCN) guidelines [8], the TKIs sorafenib, axitinib and cabozantinib are all recommended for second-line therapy, without restriction. Since there is a lack of comparative data regarding the mechanism of action of these TKIs, the present study was designed to evaluate differences in growth and angiogenic behavior of sunitinib-resistant endothelial cells after exposure to axitinib or sorafenib.

## 2. Materials and Methods

### 2.1. Human Umbilical Vein Endothelial Cells

Human umbilical vein endothelial cells (HUVECs) were selected as a model although endothelial cells isolated from renal cell carcinoma would have been more suitable. Unfortunately, endothelial RCC cells did not allow passaging more than three times and cultivation was therefore only possible for a very limited time period. This short time period was not sufficient to induce resistance to sunitinib, which was the object of the present investigation. Hence, HUVECs, which can be cultivated over longer time periods, were employed in the present investigation. The umbilical cord vessel was cannulated with a blunt needle and then perfused with phosphate buffered saline (PBS; Gibco/Invitrogen, Karlsruhe, Germany) to wash out blood. One end of the vein was then clamped with clamping scissors; the open end was infused with 5 mL dispase (Gibco/Invitrogen) and then also clamped. Following 15 min incubation at 37 °C, the clamping scissors were removed and the enzyme solution containing the HUVECs was flushed from the cord with PBS and collected. The effluent was centrifuged, the supernatant discarded and the remaining cells transferred to culture flasks in culture medium consisting of Medium 199 (M199; Biozol, Munich, Germany), 10% FCS (Gibco/Invitrogen), 10% pooled human serum (The German Red Cross Blood Donor Service, Frankfurt, Germany), 20 µg/mL endothelial cell growth factor (Boehringer, Mannheim, Germany), 0.1% heparin (Ratiopharm, Ulm, Germany), 1% GlutaMAX, 100 ng/mL gentamycin and 20 mM HEPES-buffer (pH 7.4, all: Gibco/Invitrogen). HUVECs were cultured at 37 °C in a humidified incubator with 5% CO_2_. The medium was exchanged three times a week. Each experiment employed HUVECs derived from one umbilical cord. The Institutional Ethics Committee of the Goethe-University Hospital, Frankfurt, Germany, approved the investigation and waived the need for consent, since HUVECs were anonymously used for in vitro assay with no link to patient data.

### 2.2. Dose-Response Analysis

Cell growth was evaluated using the 3-(4,5-dimethylthiazol-2-yl)-2,5-diphenyltetrazolium bromide (MTT) dye reduction assay (Roche Diagnostics, Penzberg, Germany). Five thousand HUVECs were pipetted in triplicate to 96-well plates with normal medium (control) or medium enriched with the following TKIs at different concentrations: Sunitinib (1–4 µM), sorafenib (20–100 µM), axitinib (1–4 µM). After 24, 48 and 72 h, 10 µL MTT (0.5 mg/mL) was added for an additional 4 h. Cells were then lysed in solubilization buffer (10% SDS in 0.01 M HCl) overnight at 37 °C, 5% CO_2_. Absorbance at 550 nm was assessed with a microplate enzyme-linked immunosorbent assay (ELISA) reader. To correlate absorbance with cell number, a defined number of cells ranging from 2500 to 160,000 cells/well was added to the microtiter plates (in triplicate). After subtracting the background absorbance (cell culture medium alone), results were expressed as mean cell number. IC_50_-values were determined and, once established, used for ongoing studies (2 µM sunitinib, 2 µM axitinib, 50 µM sorafenib). To exclude toxic effects of the TKIs, cell viability was determined by trypan blue (Gibco/Invitrogen).

### 2.3. Sequence Therapy

HUVECs were exposed to 2 µM sunitinib for 24 h and then subjected to the experimental analysis indicated below. In parallel, HUVECs were treated with sunitinib over 6 weeks to induce resistance. Thereafter, therapy was switched and HUVECs were then treated either with 2 µM axitinib or 50 µM sorafenib. For comparison HUVECs were also treated with sunitinib or medium without TKIs. Twenty-four hours and six weeks after the therapeutic switch, cell cultures were subjected to experimental analysis. For controls HUVECs were initially exposed to cell culture medium without sunitinib. After 6 weeks, cells were switched to axitinib, sorafenib, sunitinib or received culture medium without a TKI. Experiments were carried out 24 h after treatment start, and 24 h and 6 weeks after treatment switch (Figure 1).

### 2.4. Analysis of Cell Cycling

Cell cycle analysis was carried out on subconfluent cell cultures. HUVECs were stained with propidium iodide, using a Cycle TEST PLUS DNA Reagent Kit (BD Biosciences, Heidelberg, Germany) and then subjected to flow cytometry (FACScalibur flow cytometer, BD Biosciences). Ten thousand events were collected from each sample and data were acquired using Cell-Quest software. ModFit software (BD Biosciences) was used to assess cell cycle distribution. The number of gated cells in the G0/G1-, G2/M- or S-phase was presented as % of total cells.

### 2.5. Analysis of Cell Cycle Regulating Proteins

Cell cycle regulating proteins were examined by Western blotting. Protein lysates of HUVECs were separated on 7–12% polyacrylamide gel via electrophoresis at 100 V for 90 min. Proteins were then transferred to nitrocellulose membranes. Membranes were blocked with skim milk powder for 1 h and then incubated overnight with monoclonal antibodies directed against the following cell cycle proteins: CDK1 (IgG1, clone 1), pCDK1/Cdc2 (pY15; IgG1, clone 44/Cdk1/Cdc2), CDK2 (IgG2a, clone 55), cyclin A (IgG1, clone 25), cyclin B (IgG1, clone 18; all from BD Biosciences) and pCDK2 (Thr160; Cell Signaling, Frankfurt, Germany). AKT protein was investigated through the following antibodies: PKBα/Akt (IgG1, clone 55) and pAKT (IgG1, pS472/pS473, clone 104A282; both from BD Biosciences). HRP-conjugated goat anti-mouse IgG and HRP-conjugated goat anti-rabbit IgG (both: Cell Signaling) served as secondary antibodies. Membranes were briefly incubated with ECL detection reagent (Amersham/GE Healthcare, München, Germany) to visualize the proteins and then analyzed with Fusion FX7 apparatus (Peqlab, Erlangen, Germany). β-actin (Cell Signaling) served as the internal control. Pixel density analysis of the protein bands (both total and phosphorylated) and calculation of the ratio of protein intensity/β-actin intensity was carried out with GIMP 2.8 software.

### 2.6. Tube Formation Assay

The tube formation assay was employed to model the reorganization of angiogenesis. µ-Slides (8 well, ibidi GmbH, Gräfelfing, Germany) were coated with 10 µL Matrigel (Corning, New York, NY, USA). HUVEC were detached from the culture flask by trypsinization and resuspended in TKI-containing medium (controls were without TKI). Subsequently, 50 µL HUVEC cell suspension was transferred in triplicate to the matrigel coated wells at a concentration of 10,000 cells/well and incubated in a humidified incubator at 37 °C und 5% CO_2_. Tube formation was monitored using the IncuCyte Zoom system (Sartorius, Göttingen, Germany) by taking images of each well every 30 min for 24 h. After 5 h incubation tube length was quantified with the software “WimTube Image Analysis” (Onimagin Technologies SCA, Cordoba, Spain).

### 2.7. Scratch Wound Assay

The scratch wound assay was used to examine the horizontal migration of HUVECs. HUVECs were seeded onto 96-well ImageLock plates (Sartorius, Goettingen, Germany) at 30,000 cells/well. The plates were previously coated with 200 µL fibronectin (25 µg/mL) at 4 °C for 1 h. Cells were allowed to attach for 4 h (37 °C, 5% CO_2_) and then subjected to drug treatment as described above. Twenty-four hours later, a defined scratch of about 700 µm was made with an IncuCyte^®^ WoundMaker (Sartorius). Detached cells were removed by washing with PBS with Ca_2+_ and Mg_2+_. Cell culture medium was then renewed with 200 µL medium according to the treatment protocol. Controls received cell culture medium without TKI. Plates were incubated in Incucyte^®^ Zoom (Sartorius) at 37 °C, 5% CO_2_ and photographed every 4 h for 24 h. Each experiment was done in triplicate. Relative wound density was calculated by the software “WimScratch” (Onimagin Technologies SCA).

### 2.8. Statistics

Means +/− SD were calculated. To exclude coincidence, all experiments were repeated three to five times. Statistical significance was evaluated with Student’s *t*-test. *p* < 0.05 was considered significant.

## 3. Results

### 3.1. Dose-Response Analysis

HUVEC growth was significantly diminished when the cells were treated with sunitinib at concentrations ranging from 1–4 µM. Nearly 50% growth reduction was achieved with 2 µM sunitinib. Axitinib caused growth suppression when applied at 1–4 µM as well. Higher concentrations of sorafenib were necessary (40–100 µM) to distinctly affect the HUVECs. Half-maximum efficacy was seen at 50 µM (Figure 2A). Based on the MTT-test, all further experiments were done with 2 µM sunitinib, 2 µM axitinib and 50 µM sorafenib; 2 µM sunitinib increased the number of cells in the G0/G1-phase and simultaneously reduced the number of S-phase cells when added for 24 h (start phase; Figure 2B). The alteration in cell cycling was accompanied by diminished expression of pCDK1, pCDK2, cyclin A, cyclin B and pAkt (evaluated after 24 h; Figure 2C,D). 

### 3.2. Sequence Therapy, Switch

Following six weeks sunitinib or medium (control) exposure, sorafenib or axitinib treatment was initiated. Controls displayed diminished HUVEC growth, independent from the TKI used, whereas no effect on cell growth was seen when HUVEC had been pretreated with sunitinib for six weeks and then subjected to either sorafenib, axitinib or sunitinib for 24 h (Figure 3A).

Protein analysis demonstrated decreased pCDK1 and pAKT following 24 h sunitinib, sorafenib or axitinib exposure in the controls. Total AKT was also diminished by sorafenib and sunitinib. However, both axitinib and sorafenib elevated CDK1. Axitinib additionally reduced cyclin B, and sunitinib additionally diminished cyclin A (Figure 3B,C). Following six weeks sunitinib pretreatment, no distinct modification of cell cycle regulating proteins was seen. Axitinib, however, down-regulated pAKT and pCDK1 (but not cyclin B). Sorafenib down-regulated pAKT as well, but also acted on cyclin B and pCDK2 (Figure 3B,C). Cell cycle evaluation displayed an increase of G0/G1-phase cells and decrease of S-phase cells in the controls (sorafenib > axitinib, sunitinib) but not under TKIs following six weeks sunitinib pretreatment (Figure 3D).

### 3.3. Sequence Therapy, End

Following six weeks incubation with medium (End, medium without sunitinib) or sunitinib (End, medium with sunitinib), HUVEC were treated for a further six weeks with the TKIs and then subjected to analysis. In the control experiments (without sunitinib), neither axitinib nor sorafenib caused significant differences in cell growth, compared to cells treated with culture medium alone (Figure 4A). However, when HUVEC were pretreated with sunitinib for six weeks and then switched to a further six weeks TKI incubation, axitinib (but not sorafenib) was shown to significantly reduce HUVEC growth (Figure 4A). No effect on cell growth was seen when HUVECs were only treated with sunitinib over 12 weeks. Sunitinib elevated pCDK1 but reduced pAKT. Axitinib only diminished pAKT and CDK1 but enhanced pCDK1. Sorafenib suppressed cyclin A and CDK1, but elevated pAKT and pCDK1. When HUVECs were pretreated with sunitinib for six weeks and TKIs were then added, both sorafenib and axitinib diminished CDK1, pCDK1, cyclin A and B and pAKT (Figure 4B,C). Additionally, axitinib (but not sorafenib) reduced pCDK2. Treatment with cell culture medium for six weeks (End, medium without sunitinib) and then with axitinib for six weeks resulted in a reduction of G0/G1- and an increase in S phase cells (Figure 4D). Strong elevation of G2/M- and loss of G0/G1-phase cells were recorded in cultures treated for six weeks with sunitinib, whereas treatment with sorafenib did not induce significant alterations, compared to the controls. Exposing HUVECs to sunitinib for six weeks (End, medium with sunitinib) and then to axitinib for a further six weeks enhanced the HUVEC percentage in G0/G1 and diminished the HUVEC percentage in the S- and G2/M-phase. No alteration was seen after six weeks exposure to sorafenib. Six weeks sunitinib followed by an additional six week sunitinib incubation reduced the G0/G1- and S-phase proportion but considerably up-regulated cells in the G2/M-phase (Figure 4D). The trypan blue exclusion test done at the start, switching and end phase did not reveal signs of necrosis.

### 3.4. Tube Formation

Sunitinib, added to freshly isolated HUVECs, led to a significant reduction in tube length after 24 h incubation (Figure 5A). Treatment switch after six weeks to 24 h axitinib or sorafenib was accompanied by a reduced tube length in the control experiment (no sunitinib pretreatment), but not when HUVECs were pretreated with sunitinib for six weeks. Six weeks sunitinib plus a further 24 h sunitinib exposure even caused a significant increase in tube length, compared to the untreated cell cultures (Figure 5B). Application of TKIs for six weeks to HUVECs that had been pre-cultivated for six weeks with cell culture medium without sunitinib was associated with an increased tube length in the presence of sorafenib and sunitinib. Axitinib did not induce any alteration, compared to the untreated controls (Figure 5C). In contrast, when HUVECs were pretreated with sunitinib for six weeks and then switched to six weeks axitinib, sorafenib or (further six weeks) sunitinib, both axitinib and sorafenib (but not sunitinib) significantly diminished tube length (axitinib > sorafenib).

### 3.5. Wound Closure

Sunitinib significantly inhibited wound closure during the start phase (Figure 6A). Wound closure was also inhibited when HUVECs were treated with culture medium for six weeks and then exposed to either sunitinib, axitinib or sorafenib for 24 h (switch). Figure 6B shows values 8 h after setting the scratch. All three TKIs, sunitinib, axitinib and sorafenib, slowed HUVEC wound closure when added for 24 h, although the effect was not as strong as seen under control conditions. Exposing HUVECs to cell culture medium for six weeks followed by a further six weeks TKI treatment (End) prevented wound closure regardless of the TKI applied (axitinib, sunitinib > sorafenib) (Figure 6C). Differences were seen when HUVECs were pretreated with sunitinib for six weeks and then exposed to the TKIs for a further six weeks. Then all three TKIs slowed wound closure, with sorafenib being more effective than axitinib or sunitinib (Figure 6C).

## 4. Discussion

This study was designed to investigate how two second line TKIs, axitinib and sorafenib, affect endothelial cell growth and angiogenic behavior following first line TKI failure with sunitinib. The clinical relevance is not only restricted to treatment where first-line therapy is based exclusively on a TKI, since combining TKIs with immune ICIs has been approved as a new first-line option to treat RCC. However, systematic reviews and meta-analyses have shown that the majority of patients are unlikely to benefit from ICIs [9], and resistance develops whether the patients have been treated with a TKI monotherapy or a TKI-ICI combination [10]. TKIs have therefore also been suggested as second-line therapy for RCC patients, refractory to ICIs [7,10]. The EAU, ESMO and NCCN recommend any TKI, including axitinib and sorafenib, that has not been used first line [10], as do others [11]. Accordingly, Deuker et al. has not seen major differences in the efficacy of several TKIs after an immunotherapy-based combination regimen [12]. Overall survival of patients treated with axitinib or sorafenib, subsequent to discontinuing an ICI-TKI regimen or sunitinib therapy, was similar [13,14]. In contrast, others have attested to better results with axitinib, instead of sorafenib, as a second line option [7,15]. Both Schmidinger et al. and Géczi et al. have recommended second-line axitinib post sunitinib [16,17], whereas sorafenib as a second-line treatment (following sunitinib failure) has been favored by other investigators [18].

The mechanisms underlying sunitinib resistance are multifacetted and not completely understood. Alteration of the noncoding RNAs expression level, upregulation of pro-angiogenic signaling, the RAF/MEK/ERK and/or the PI3K/AKT/mTOR pathway are all considered to be resistance factors. We did not concentrate on this issue. However, diminished expression of pCDK1, pCDK2, cyclin A and pAkt seen under short term sunitinib exposure was not seen under long term sunitinib treatment. Therefore, these cell cycle regulating proteins might be (at least in part) responsible for resistance acquisition in our model.

Axitinib, but not sorafenib, significantly reduced the HUVEC cell number when given chronically over six weeks to sunitinib-resistant cells. Short-term axitinib exposure for 24 h did not suppress HUVEC growth, indicating that axitinib may not rapidly overcome sunitinib resistance and long-term application may be necessary to initiate axitinib effectivity. It should be noted in this context that cell growth data are all based on the MTT assay, which serves as a well-established method to determine cell viability and number. Nevertheless, mitochondrial hyperactivation and increased MTT reduction, as has recently been observed when tumor cells have been radiated [19], may have occurred. Given that the tumor cells’ metabolic activity may also be elevated following TKI exposure, the cytotoxic potential of the drugs could be underestimated. This issue has not been dealt with in the present investigation, and therefore remains speculative.

Distinct alterations of cell signaling proteins under axitinib have been seen only after six weeks. Axitinib then strongly blocked CDK1 (both total and activated), cyclin A and B and pAKT, whereas only pCDK1 (slightly) and pAKT were modified in HUVECs after 24 h axitinib exposure. Loss of Ki-67 along with p27 down-regulation has been observed when HUVECs are exposed to axitinib [20]. Reduction of endothelial pAKT by axitinib has been shown by others to be responsible for proliferation suppression [21]. No data are available with respect to the cyclin-CDK-axis. However, axitinib increased p21 in glioma cells [22]. This is important, since both p21 and p27 serve as prominent cell cycle regulators by decreasing CDK activity and reducing the cyclin expression level [23,24].

Consequently, it may be assumed that axitinib counteracts sunitinib resistance by deactivating AKT and cyclin-CDK signaling. Nevertheless, when assessing axitinib’s mode of action, it should be kept in mind that sorafenib acted on CDK-cyclin and AKT even more strongly than axitinib did, but without suppressing HUVEC growth. Since sorafenib did not influence cell cycle progression, the protein alterations seen under sorafenib may be of limited relevance to cell growth regulation. In a hepatocellular carcinoma model, sorafenib has been documented to inhibit HUVEC growth by only 20%, but to exert a 75% blocking activity in the scratch assay and a 50% blocking activity in tube formation. The same sorafenib concentration completely abolished pAKT [25]. These observations accord with our hypothesis that sorafenib inhibits the angiogenic behavior of HUVECs rather than their growth and that suppression of pAKT may be the relevant factor triggering antiangiogenic behavior. While sorafenib did not influence HUVEC growth in our experimental model, evidence has been presented indicating that sorafenib stops wound closure and tube formation. In our scratch assay, sorafenib was superior to axitinib when HUVECs were continuously exposed to the drugs for six weeks. Presumably, sorafenib primarily acts as an angiogenesis regulator via the AKT pathway, thereby counteracting acquired resistance towards sunitinib. While this requires further verification, recent investigation on a hepatoma and breast cancer model point to the importance of AKT as an angiogenesis driver and the relevance of sorafenib in preventing HUVEC migration and capillary tube formation by dephosphorylating AKT [26,27].

The degree to which cyclin family members are involved in angiogenesis is not yet clear. However, multi-kinase inhibitors targeting specific CDKs have already been introduced, demonstrating potent reduction of endothelial cell migration and tube formation in vitro and in vivo [28,29]. The strong effects of sorafenib on CDKs and cyclins should, therefore, be interpreted in the context of antiangiogenic activity, whereby the double function of axitinib on angiogenesis and growth behavior should be kept in mind.

We did not analyze apoptosis in the present project and, therefore, cannot comment on whether axitinib or sorafenib are involved in these processes and whether they may activate the apoptotic cascade in sunitinib-resistant cells. Axitinib did not induce apoptosis in endothelial cells in vitro and in fibrosarcoma and melanoma bearing mice [30], whereas sorafenib forced the expression of cleaved PARP-1 and caspase-3 in HUVECs and elevated the proportion of cells in the sub-G1 fraction [31,32]. Whether sorafenib exerts this activity under sunitinib-resistance as well, is not clear and requires further evaluation.

Both axitinib and sorafenib exhibit antitumor properties that, though not the same, can provide advantage as second line drugs. Since the mechanisms of action of axitinib and sorafenib differ from that of sunitinib, it may be assumed that differences in the molecular mode of action of both TKIs, compared to those of sunitinib, are responsible for counteracting negative feedback loops caused by sunitinib. Sunitinib acts on the vascular endothelial growth factor receptor (VEGFR) 1, 2, 3, the platelet-derived growth factor receptor (PDGFR) α and β, c-KIT, FLT-3 and RET. Sorafenib does not influence FLT-3, but Raf instead, and axitinib exclusively modulates PDGFR α, but not β, and does not alter FLT-3 and RET signaling [33]. Chronic blockage of VEGFR-signaling has been associated with activation of the mTOR-pathway. In the current study evidence is provided that both axitinib and sorafenib down-regulate pAKT equally well in sunitinib-resistant HUVECs.

Cabozantinib, an oral TKI targeting VEGFR2, MET, AXL, RET, KIT and FLT3 that was approved by the FDA in 2016 for patients with advanced RCC, who had formerly been treated with one or more antiangiogenic drugs [34], was not investigated in the current study. Sorafenib, axitinib and cabozantinib are all recommended in EAU, ESMO and NCCN guidelines [8] for second-line therapy. However, no statistically significant difference among the three drugs is apparent with regard to progression free survival [35]. An animal study has revealed blockage of angiogenesis under cabozantinib, whilst tumors become more infiltrative and escape treatment [36]. These findings point to a mechanism similar to the one shown for sorafenib in counteracting sunitinib resistance by interfering with pathways connected to angiogenesis, but not to cell growth. Considering that cabozantinib along with axitinib and sorafenib are recommended as second line treatment, ongoing studies should be directed towards comparing the efficacy of all three drugs.

Limitations of the present investigation should be considered. Our HUVEC model may not perfectly reflect endothelial cells derived from the kidney, making ongoing experiments with further endothelial cell types necessary. We also did not investigate apoptosis in our experiments and, therefore, cannot judge whether selective apoptotic pressure may have played a role in the effectiveness of sunitinib, sorafenib and axitinib. A study employing T-cells has shown that all three drugs induce apoptosis, however by different mechanisms with unique features of axitinib, when compared to sunitinib and sorafenib [37]. Respective data in regard to endothelial cells are not available. Whether this mechanistic difference may provide an advantage of axitinib over sorafenib in terms of apoptosis induction therefore remains a prospective area of investigation.

## 5. Conclusions

Evidence is provided that both axitinib and sorafenib might be equally well qualified as second-line treatment options following sunitinib failure. These TKIs differ with respect to their mode of action, with axitinib inhibiting angiogenesis and growth, while sorafenib predominantly inhibits migration and tube formation. These in vitro experiments should now be followed by in vivo studies to facilitate clear clinical guidelines.

## Figures and Tables

**Figure 1 biomedicines-09-01630-f001:**
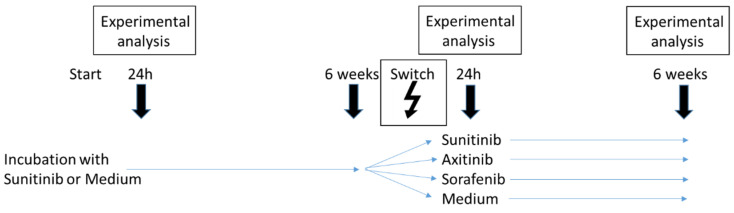
Schematic illustration of the study protocol.

**Figure 2 biomedicines-09-01630-f002:**
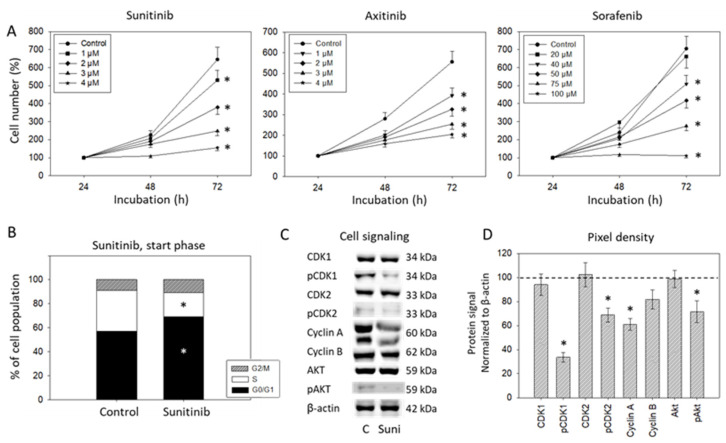
(**A**) Dose-response analysis. HUVECs were incubated with different concentrations of sunitinib, axitinib or sorafenib, and cell number was evaluated after 24 (100%), 48 and 72 h by MTT assay. Means and standard deviation are indicated, n = 3, * = *p* ≤ 0.05. (**B**) Influence of 2 µM sunitinib on G0/G1-, S- and G2/M-phases of the cell cycle in HUVEC after 24 h (n = 3). * indicates significant difference to untreated controls. (**C**) Protein profile of cell-cycle-regulating proteins after exposure to 2 µM sunitinib (Suni). Controls (**C**) were untreated (0 µM sunitinib). One representative of three separate experiments is shown. Each protein analysis was accompanied by a β-actin loading control. One representative internal control is shown. (**D**) The ratio of protein intensity/β-actin intensity was calculated and expressed as a percentage of the controls, set to 100%. * indicates significant difference to controls, *p* ≤ 0.05. n = 3.

**Figure 3 biomedicines-09-01630-f003:**
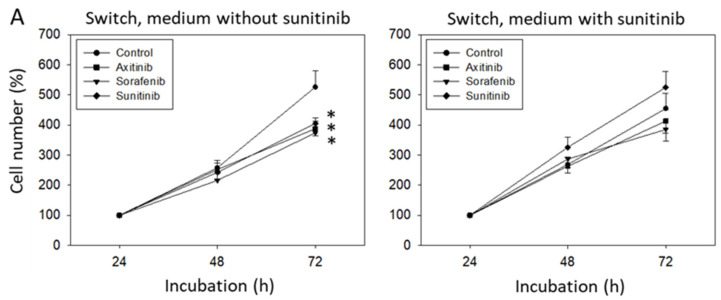
(**A**) Cell growth analysis by the MTT assay. HUVECs were incubated with sunitinib for 6 weeks and then switched to axitinib or sorafenib for 24 h (Switch, medium with sunitinib). In parallel, HUVECs were incubated with sunitinib for 6 weeks and incubated for a further 24 h with sunitinib (Sunitinib) or with culture medium alone (Control). Figure 3A, “Switch, medium without sunitinib” indicates cell growth behavior of HUVECs incubated with medium alone for 6 weeks and then switched to axitinib or sorafenib for 24 h. In parallel, HUVECs were incubated with cell culture medium for 6 weeks and then incubated for a further 24 h with sunitinib (Sunitinib) or with culture medium alone (Control). * indicates significant difference to untreated controls (n = 6). (**B**) Protein profile of cell-cycle-regulating proteins after 6 weeks sunitinib (medium with sunitinib) or cell culture medium alone (medium without sunitinib) followed by a 24 h switch to axitinib (Axi), sorafenib (Sora), to further 24 h sunitinib (Suni) or 24 h culture medium alone (**C**). Each protein analysis was accompanied by a β-actin loading control. One representative internal control is shown. (**C**) The ratio of protein intensity/β-actin intensity was calculated and expressed as a percentage of the controls, set to 100%. * indicates significant difference to controls, *p* ≤ 0.05. n = 3. (**D**) Cell cycle analysis after 6 weeks incubation with sunitinib (Switch, medium with sunitinib) or cell culture medium alone (Switch, medium without sunitinib) and subsequent 24 h incubation with axitinib, sorafenib, sunitinib or culture medium alone (**C**) (n = 3). * indicates significant difference to the untreated controls.

**Figure 4 biomedicines-09-01630-f004:**
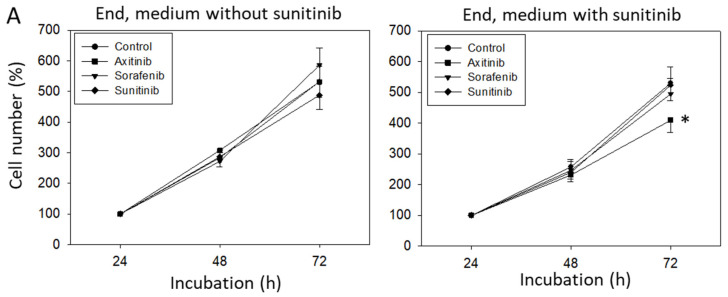
(**A**) Cell growth analysis by the MTT assay. HUVECs were incubated with sunitinib for 6 weeks and then switched to axitinib or sorafenib for a further 6 weeks (End, medium with sunitinib). In parallel, HUVECs were incubated with sunitinib for 6 weeks and incubated for a further 6 weeks with sunitinib (Sunitinib) or with culture medium alone (Control). Figure 4A, “End, medium without sunitinib” indicates cell growth behavior of HUVECs incubated with medium alone for 6 weeks and then switched to axitinib or sorafenib for 6 weeks. In parallel, HUVECs were incubated with cell culture medium for 6 weeks and then incubated for a further 6 weeks with sunitinib (Sunitinib) or with culture medium alone (Control). * indicates significant difference to untreated controls (n = 6). (**B**) Protein profile of cell-cycle-regulating proteins after 6 weeks sunitinib (Medium with sunitinib) or cell culture medium alone (Medium without sunitinib) followed by a 6 week switch to axitinib (Axi), sorafenib (Sora), to a further 6 weeks sunitinib (Suni) or 6 weeks culture medium alone (**C**). Each protein analysis was accompanied by a β-actin loading control. One representative internal control is shown. (**C**) The ratio of protein intensity/β-actin intensity was calculated and expressed as a percentage of the controls, set to 100%. * indicates significant difference to controls, *p* ≤ 0.05. n = 3. (**D**) Cell cycle analysis after 6 weeks incubation with sunitinib (End, medium with sunitinib) or cell culture medium alone (End, medium without sunitinib) and subsequent 6 week incubation with axitinib, sorafenib, sunitinib or culture medium alone (**C**) (n = 3). * indicates significant difference to untreated controls.

**Figure 5 biomedicines-09-01630-f005:**
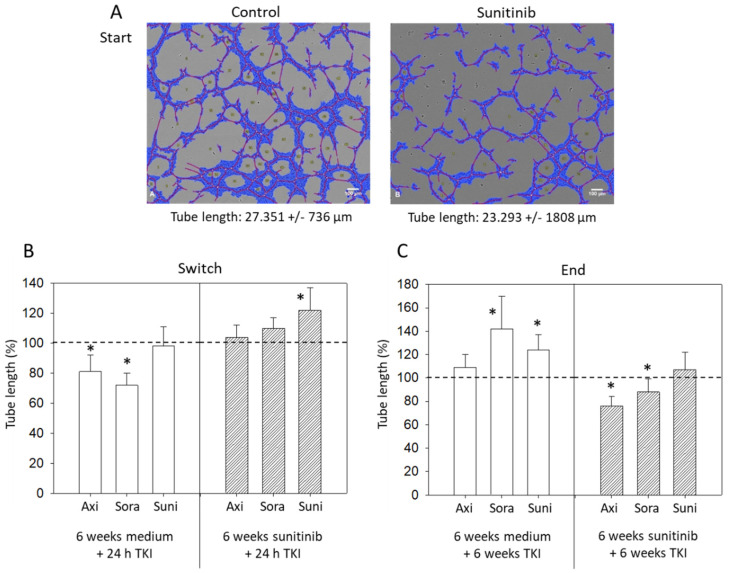
(**A**) Tube formation after 24 h sunitinib exposure, compared to untreated controls. One representative figure and the mean tube length of treated versus untreated HUVECs are shown (n = 3). Blue = cell covered area, red = tubes, yellow = branching points. (**B**) Evaluation of tube length after 6 weeks sunitinib (6 weeks sunitinib + 24 h TKI) or cell culture medium (6 weeks medium + 24 h TKI) followed by 24 h axitinib, sorafenib or sunitinib (Switch). Untreated controls were set to 100%. * indicates significant difference to the untreated controls. (**C**) Evaluation of tube length after 6 weeks sunitinib (6 weeks sunitinib + 6 weeks TKI) or cell culture medium (6 weeks medium + 6 weeks TKI) followed by 6 weeks axitinib, sorafenib or sunitinib (End). Untreated controls were set to 100%. * indicates significant difference to untreated controls.

**Figure 6 biomedicines-09-01630-f006:**
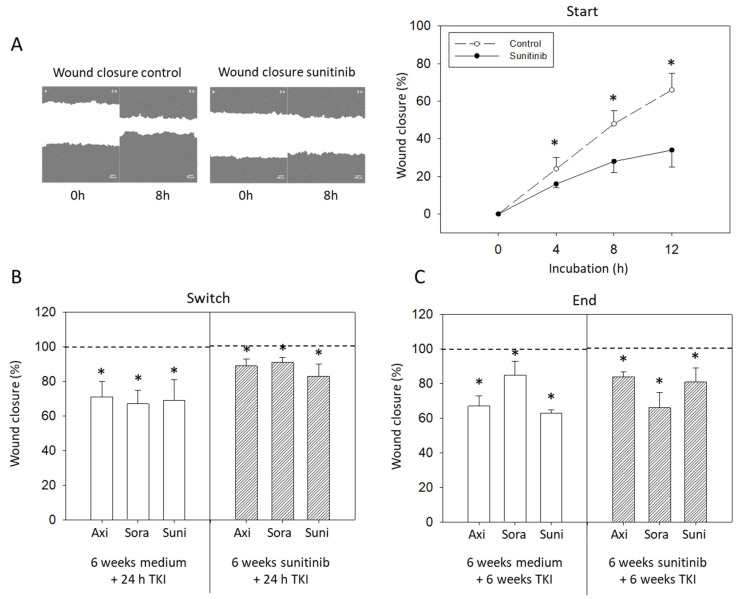
(**A**) Wound closure analyzed after 24 h sunitinib exposure and compared to untreated controls. One representative figure and the mean wound closure of treated versus untreated HUVECs are shown (n = 3). (**B**) Evaluation of wound closure after 6 weeks sunitinib (6 weeks sunitinib + 24 h TKI) or cell culture medium (6 weeks medium + 24 h TKI) followed by 24 h axitinib, sorafenib or sunitinib (Switch). Untreated controls were set to 100%. * indicates significant difference to the untreated controls. (**C**) Evaluation of wound closure after 6 weeks sunitinib (6 weeks sunitinib + 6 weeks TKI) or cell culture medium (6 weeks medium + 6 weeks TKI) followed by 6 weeks axitinib, sorafenib or sunitinib (End). Untreated controls were set to 100%. * indicates significant difference to the untreated controls.

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
