# Peer review of "Antiangiogenic Properties of Axitinib versus Sorafenib Following Sunitinib Resistance in Human Endothelial Cells—A View towards Second Line Renal Cell Carcinoma Treatment"

_biomedicines, 2021, doi:10.3390/biomedicines9111630_

Round 1
Reviewer 1 Report
Juengel et al., investigated and compared the anti-angiogenic properties of the different tyrosine kinase inhibitors (TKIs) using Human umbilical vein endothelial cells. The study highlighted that axitinib and sorafenib reduced HUVEC tube length and prevented wound closure when applied to sunitinib resistant HUVECs for 6 weeks and both axitinib and sorafenib might be equally well qualified as second-line treatment option following sunitinib failure.
Introduction needs details on the brief overview on immune checkpoint inhibitors (ICI) and different TKIs with respect to treatment of renal carcinomas. Describe how the details on the isolation procedure of HUVEC cells (Supplementary file). Did the author isolated the HUVEC cells in batches of cells for the all the experiments? Have authors examined the HUVEC cell viability at the before beginning of the incubation with TKIs and at the end of 6-weeks before switch and during the course of the 24h to 6-weeks?
In my opinion, it is important to perform densitometric analysis of the all the Immunoblots presented in the manuscript and then analyze the data on the axitnib and sorafenib with sunitinib comparing to non-treated controls in AKT and cyclin-CDK signaling.
The figure 5A needs extensive description on the methods and staining of the immunocytochemistry. Did authors carry out the statistical analysis of the figure 5A?
With respect to statistical analysis, I believe two-way ANOVA for the data on tube length and wound closure will provide better statistical interpretations.
I noted minor typographical in the manuscript.
Author Response
Comment 1: Introduction needs details on the brief overview on immune checkpoint inhibitors (ICI) and different TKIs with respect to treatment of renal carcinomas.
Our answer: We have added to the “Introduction”: “Multiple tyrosine kinase inhibitors (TKIs), acting on specific growth factor receptors or drugs interfering with the mechanistic target of rapamycin (mTOR) pathway have revealed promise in terms of progression free and overall survival [3]. (line 39) Sorafenib was the first multi-targeted TKI approved for first-line treatment of metastatic RCC, which was then replaced by sunitinib and pazopanib as the standard of care. Recently, the NCCN Guidelines has recommended axitinib as first-line option for clear cell RCC with favorable and poor/intermediate risk and cabozantinib for patients with poor/intermediate risk. The combination of lenvatinib and the mTOR-inhibitor, everolimus, is recommended for the subsequent therapy of clear cell RCC and systemic therapy for non-clear cell RCC [4].”.
“Novel concepts concentrating on immune checkpoint inhibitors (ICI), (line 48) targeting programmed death 1 (PD-1), the PD-Ligand 1 (PD-L1) and cytotoxic T-lymphocyte–associated antigen 4 (CTLA-4), are meanwhile under investigation. ICIs include nivolumab (anti–PD-1), pembrolizumab (anti–PD-1), avelumab (anti–PD-L1), atezolizumab (anti–PD-L1), and ipilimumab (anti–CTLA-4) [5]”.
References have been included as well.
Comment 2: Describe how the details on the isolation procedure of HUVEC cells (Supplementary file). Did the author isolated the HUVEC cells in batches of cells for the all the experiments?
Our answer: We have now provided more information about the HUVEC isolation protocol and why we used HUVECs instead of endothelial RCC cells (this was a rightful concern of referee 2). Materials and Methods, Human umbilical vein endothelial cells, now reads (line 65): “Human umbilical vein endothelial cells (HUVECs) were selected as a model although endothelial cells isolated from renal cell carcinoma would have been more suitable. Unfortunately, endothelial RCC cells did not allow passaging for more than three times and cultivation was therefore only possible for a very limited time period. This short time period was not sufficient to induce resistance to sunitinib, which was the object of the present investigation. Hence, HUVECs, which can be cultivated over longer time periods, were employed in the present investigation. The umbilical cord vessel was cannulated with a blunt needle and then perfused with phosphate buffered saline (PBS; Gibco/Invitrogen; Karlsruhe, Germany) to wash out blood. One end of the vein was then clamped with a clamping scissors, the open end was infused with 5 ml dispase (Gibco/Invitrogen) and then also clamped. Following 15 min incubation at 37°C, the clamping scissors were removed and the enzyme solution containing the HUVECs was flushed from the cord with PBS and collected. The effluent was centrifuged, the supernatant discarded and the remaining cells transferred to culture flasks in culture medium consisting of Medium 199 (M199; Biozol, Munich, Germany), 10% FCS (Gib-co/Invitrogen), 10% pooled human serum (The German Red Cross Blood Donor Service, Frankfurt, Germany), 20 µg/ml endothelial cell growth factor (Boehringer, Mannheim, Germany), 0.1% heparin (Ratiopharm, Ulm, Germany), 1% GlutaMAX, 100 ng/ml gentamycin and 20 mM HEPES-buffer (pH 7.4, all: Gibco/Invitrogen). HUVECs were cultured at 37°C in a humidified incubator with 5% CO2. Medium change was done three times a week. Each experiment employed HUVECs derived from one umbilical cord”.
Comment 3: Have authors examined the HUVEC cell viability at the before beginning of the incubation with TKIs and at the end of 6-weeks before switch and during the course of the 24h to 6-weeks?
Our answer: Cell viability was routinely controlled by the trypan blue exclusion test. We apologize for forgetting to include this into the manuscript. Materials and Methods, Dose-response-analysis, last sentence, now reads (line 103): “To exclude toxic effects of the TKIs, cell viability was determined by trypan blue (Gibco/Invitrogen)”.
We also added to Results, Sequence therapy, end: “No alteration was seen after 6 weeks exposure to sorafenib. 6 weeks sunitinib followed by an additional 6 week sunitinib incubation reduced the G0/G1- and S-phase proportion but considerably up-regulated cells in the G2/M-phase (fig. 4C). (line 264) The trypan blue exclusion test done at the start, switching and end phase did not reveal signs of necrosis”.
Comment 4: In my opinion, it is important to perform densitometric analysis of the all the Immunoblots presented in the manuscript and then analyze the data on the axitinib and sorafenib with sunitinib comparing to non-treated controls in AKT and cyclin-CDK signaling.
Our answer: Pixel density analysis has been done. We included in Materials and Methods, section “Analysis of cell cycle regulating proteins” (line 145): “Pixel density analysis of the protein bands (both total and phosphorylated) and calculation of the ratio of protein intensity/β-actin intensity was carried out with GIMP 2.8 software.”
The appropriate data are now shown in figures 2, 3, 4. Figure legends have been changed accordingly.
Comment 5: The figure 5A needs extensive description on the methods and staining of the immunocytochemistry. Did authors carry out the statistical analysis of the figure 5A?
Our answer:
In fact, the difference in tube length was significant. We would like to refer to the first sentence of the section “Tube formation” in Results, which reads (line 290): “Sunitinib, added to freshly isolated HUVEC, led to a significant reduction in tube length after 24 h incubation (fig. 5A)”.
After critically rereading our manuscript, we came to the conclusion that the tube formation assay was not clearly explained. The respective section in Materials and Methods, Tube formation assay, now reads: “The tube formation assay was employed to model the reorganization of angiogenesis. µ-Slides (8 well, ibidi GmbH, Gräfelfing, Germany) were coated with 10 µl Matrigel (Corning, New York, NY, USA). HUVEC were detached from the culture flask by trypsinization and resuspended in TKI-containing medium (controls were without TKI). (line 154) Subsequently, 50 µl HUVEC cell suspension was transferred in triplicate to the matrigel coated wells at a concentration of 10,000 cells/well and incubated in a humidified incubator at 37°C und 5% CO2. Tube formation was monitored using the IncuCyte Zoom system (Sartorius, Göttingen, Germany) by taking images of each well every 30 min for 24 h. After 5 h incubation tube length was quantified with the software „WimTube Image Analysis“ (Onimagin Technologies SCA, Cordoba, Spain).
Figure legend 5A now reads (line 308): “Figure 5: 5A: Tube formation after 24 h sunitinib exposure, compared to untreated controls. One representative figure and the mean tube length of treated versus untreated HUVECs are shown (n=3). Blue = cell covered area, red = tubes, yellow = branching points”.
Comment 6: With respect to statistical analysis, I believe two-way ANOVA for the data on tube length and wound closure will provide better statistical interpretations.
Our answer: We extensively discussed this comment with our statistician. Two-way ANOVA is certainly well established for analyzing datasets. Still, the data sets of the tube length and the wound closure experiment both provide independent variables. Hence, the one-way ANOVA or the t-test can serve as statistical methods of choice, yield the same p-value and, therefore, can be used equivalently. We, therefore, believe that the t-test we employed satisfactorily reflects the statistical correlation.
Comment 7: I noted minor typographical errors in the manuscript.
Our answer: The manuscript has been checked again by an English native speaker.

Reviewer 2 Report
In their paper Juengel et al. compared the cytostatic and anti-angiogenic effects of sorafenib and axitinib on human endothelial cells (HUVECs) of acquired resistance towards the sunitinib. This model mimics the sequential exposure of endothelium to the drugs that are commonly used in the chemotherapy of renal carcinoma. Using this remarkably elegant experimental approach the Authors showed some differences in the responsiveness of sunitinib-resistant HUVECs to both analysed agents that may explain failures of renal carcinoma therapy. The study is interesting; however several issues should be addressed by the Authors before its further processing:
- Are HUVECs relevant model of renal endothelium? Perhaps the Authors have the data on the reactivity of kidney endothelium?
- What is the mechanism of sunitinib-resistance of HUVECs? Is it the drug-efflux? If so the Authors could determine or at least discuss the possible fate of sorafenib and axitinib in resistant cells;
- It is unclear why the Authors undertook rather complicated protocol of MTT calibration instead of simply counting the cells. MTT test determines the metabolic activity of the cells, which can change in different conditions. Then, the Authors show differences in cell cycle distribution after drug application. Because metabolic activity and cell sizes change throughout cell cycle phases, MTT/cell number ratio estimated for control conditions may not be relevant for the drug-treated samples. I strongly recommend more traditional cell growth analyses (cell counting by: Coulter, Buerker chamber or flow-cytometry) to confirm the conclusions;
- It would also be nice to see the data on apoptosis, because they could give the information o the selective pressure exerted by the drugs on HUVECs.
Author Response
Comment 1: Are HUVECs relevant model of renal endothelium? Perhaps the Authors have the data on the reactivity of kidney endothelium?
Our answer: It is correct that endothelial cells derived from renal vessels would provide the most relevant cell type. Unfortunately, endothelial cells isolated from renal cell carcinoma (PeloBiotech, Martinsried, Germany) only allowed passaging for a maximum of three times and cultivation for a very limited time period. Therefore, we were not able to conduct our study with this cell type. This is explained in “Materials and Methods” (line 65) Human umbilical vein endothelial cells (HUVECs) were selected as a model although endothelial cells isolated from renal cell carcinoma would have been more suitable. Unfortunately, endothelial RCC cells did not allow passaging for more than three times and cultivation was therefore only possible for a very limited time period. This short time period was not sufficient to induce resistance to sunitinib, which was the object of the present investigation. Hence, HUVECs, which can be cultivated over longer time periods, were employed in the present investigation.
This aspect is also now discussed as a limitation in the last paragraph of the discussion (line 447). Limitations of the present investigation should be considered. Our HUVEC model may not perfectly reflect endothelial cells derived from the kidney, making ongoing experiments with further endothelial cell types necessary.
Comment 2: What is the mechanism of sunitinib-resistance of HUVECs? Is it the drug-efflux? If so the Authors could determine or at least discuss the possible fate of sorafenib and axitinib in resistant cells.
Our answer: The mechanisms underlying sunitinib resistance are multifacetted and not completely understood. Based on the literature, drug-efflux does not seem to be the main mechanism of TKI caused non-reponsivenes (unlike that with chemotherapeutics). Rather noncoding RNA-mediated resistance, upregulation of pro-angiogenic signaling pathways, activation of the RAF/MEK/ERK and the PI3K/AKT/mTOR pathway are considered to be responsible for non-responsiveness, along with abnormal intracellular pharmacokinetics and alterations mediated by the tumor hypoxic microenvironment. Recent studies point to the restoration of an alternative angiogenesis pathway in modulating resistance, whereas others have observed inhibition of autophagy under chronic sunitinib application.
We did not concentrate on this issue. However, diminished expression of pCDK1, pCDK2, cyclin A, and pAkt, seen under short term sunitinib exposure, was not seen long term. Therefore, these cell cycle regulating proteins might at (least in part) be responsible for the sunitinib-resistance in our model. We have now added this concept to the discussion section, second paragraph (line 361): “The mechanisms underlying sunitinib resistance are multifacetted and not completely understood. Alteration of the noncoding RNAs expression level, upregulation of pro-angiogenic signaling, the RAF/MEK/ERK and/or the PI3K/AKT/mTOR pathway are all considered to be resistance factors. We did not concentrate on this issue. However, diminished expression of pCDK1, pCDK2, cyclin A, and pAkt seen under short term sunitinib exposure was not seen under long term sunitinib treatment. Therefore, these cell cycle regulating proteins might be (at least in part) responsible for resistance acquisition in our model.”
Comment 3: It is unclear why the Authors undertook rather complicated protocol of MTT calibration instead of simply counting the cells. MTT test determines the metabolic activity of the cells, which can change in different conditions. Then, the Authors show differences in cell cycle distribution after drug application. Because metabolic activity and cell sizes change throughout cell cycle phases, MTT/cell number ratio estimated for control conditions may not be relevant for the drug-treated samples. I strongly recommend more traditional cell growth analyses (cell counting by: Coulter, Buerker chamber or flow-cytometry) to confirm the conclusions;
Our answer: We agree that the MTT assay may have limitations. However, the MTT assay is a well-established method which allows a high throughput measurement of cell number. It also serves as a routine test to evaluate drug responsiveness. In fact, formazan-based test systems (MTT, MTS, WST-1) are recommended by the International Organization for Standardization (ISO) to determine cell number and viability (ISO 10993-5:2009; ISO 19007:2018(en)). We also used a cell counter. However, this system has disadvantages as well, since cell size and shape must initially be defined, with the consequence that rounded particles may also be counted and that “oversize” cells may not be counted. Buerker or Neubauer chambers (which we use in our lab) can be employed, but it is not the method of choice when a great number of samples need analysis (as was the case in the present investigation). Furthermore, we have observed that cell clusters may drift off from the corner squares, which will not be counted. A short summary of advantages/disadvantages of endothelial cell counting techniques is given by Rahman et al. (An Overview of In Vitro, In Vivo, and Computational Techniques for Cancer-Associated Angiogenesis Studies. Biomed Res Int. 2020;2020:8857428.). The complete study has meanwhile been finalized and we are not able to verify the MTT-data by further tests done in parallel. Still, it is important to respect the comment of the referee. We, therefore decided to add to the discussion: “Short-term axitinib exposure for 24 h did not suppress HUVEC growth, indicating that axitinib may not rapidly overcome sunitinib resistance and long-term application may be necessary to initiate axitinib effectivity (line 373). It should be noted in this context that cell growth data are all based on the MTT assay, which serves as a well-established method to determine cell viability and number. Nevertheless, mitochondrial hyperactivation and increased MTT reduction, as has recently been observed when tumor cells have been radiated [19], may have occurred. Given that the tumor cells’ metabolic activity may also be elevated following TKI exposure, the cytotoxic potential of the drugs could be underestimated. This issue has not been dealt with in the present investigation, and therefore remains speculative.
Comment 4: It would also be nice to see the data on apoptosis, because they could give the information on the selective pressure exerted by the drugs on HUVECs,
Our answer: Regulation of apoptosis by sunitinib, sorafenbib, or axitinib has already been documented. Sunitinib resistance is also shown to reduce apoptosis in renal cancer cell lines (Meng et al. Dysregulation of the Sirt5/IDH2 axis contributes to sunitinib resistance in human renal cancer cells. FEBS Open Bio. 2021;11:921-931). However, to our knowledge, data concerning apoptosis of TKI-resistant endothelial cells have not been published yet. We also did not investigate apoptosis in our experiments and, therefore, cannot comment on whether there might be selective pressure during the sequential switch. An elegant study done on T-cells has presented evidence that sunitinib, sorafenib, and axitinib all induce apoptosis, however by different mechanisms, with unique features of axitinib, when compared to sunitinib and sorafenib (Stehle et al. Reduced immunosuppressive properties of axitinib in comparison with other tyrosine kinase inhibitors. J Biol Chem. 2013;288:16334-16347.). Respective data for endothelial cells are not available. Whether this mechanistic difference provides an advantage of axitinib over sorafenib in terms of apoptosis induction remains, therefore, purely speculative.
To address this aspect, we have included in the last paragraph of the discussion in regard to limitations of the present investigation (line 449): “We also did not investigate apoptosis in our experiments and, therefore, cannot judge whether selective apoptotic pressure may have played a role in the effectiveness of sunitinib, sorafenib, and axitinib. A study employing T-cells has shown that all three drugs induce apoptosis, however by different mechanisms with unique features of axitinib, when compared to sunitinib and sorafenib [37]. Respective data in regard to endothelial cells are not available. Whether this mechanistic difference may provide an advantage of axitinib over sorafenib in terms of apoptosis induction therefore remains a prospective area of investigation.”

Round 2
Reviewer 1 Report
Please correct the quotation marks from Line 158 to 159 „WimTube Image
Analysis“ to "WimTube Image Analysis"
Author Response
„WimTube Image Analysis“ reads now "WimTube Image Analysis".
Reviewer 2 Report
I have no more comments
Author Response
We are thankful that the referee is satisfied by the corrections made.